# Changes in Transcriptomic Profiles in Different Reproductive Periods in Yaks

**DOI:** 10.3390/biology10121229

**Published:** 2021-11-25

**Authors:** Shaoke Guo, Mengli Cao, Xingdong Wang, Lin Xiong, Xiaoyun Wu, Pengjia Bao, Min Chu, Chunnian Liang, Ping Yan, Jie Pei, Xian Guo

**Affiliations:** Key Laboratory of Yak Breeding Engineering of Gansu Province, Lanzhou Institute of Husbandry and Pharmaceutical Sciences, Chinese Academy of Agricultural Sciences, Lanzhou 730050, China; 82101196170@caas.cn (S.G.); 82101205332@caas.cn (M.C.); 82101211210@caas.cn (X.W.); xionglin@caas.cn (L.X.); wuxiaoyun@caas.cn (X.W.); baopengjia@caas.cn (P.B.); chumin@caas.cn (M.C.); liangchunnian@caas.cn (C.L.); pingyan63@126.com (P.Y.)

**Keywords:** yak, ovary, anestrus, estrus, pregnancy

## Abstract

**Simple Summary:**

The molecular regulation mechanism of yak ovarian activity has attracted extensive attention. This study investigated the global gene expression profiles in different reproductive stages (anestrus, estrus, and pregnancy) by RNA-seq technology. Enrichment analysis revealed that DEGs were involved in the process of follicular growth, ovulation, and hormone metabolism. This study explored the regulation mechanism of the yak ovary in the reproductive cycle and laid a theoretical foundation for further understanding the reproductive characteristics of yak.

**Abstract:**

Yak reproductive characteristics have received extensive attention, though the molecular regulation mechanism of its ovarian activity remains to be explored. Therefore, this study initially conducted a comparative analysis of yak ovarian activities in anestrus, estrus, and pregnancy regarding their morphology and histology, followed by implementing RNA sequencing (RNA-seq) technology to detect the overall gene expression and biological mechanism in different reproductive stages. H&E staining showed that there were more growing follicles and mature follicles in ovarian tissue sections during estrus than ovarian tissues during non-estrus. The RNA-seq analysis of yak ovary tissues in three periods showed that DEGs related to follicular development and hormone metabolism were screened in the three comparison groups, such as *COL1A2*, *NR4A1*, *THBS2*, *PTGS2*, *SCARB1*, *STAR*, and *WNT2B*. Bioinformatics analysis showed that these DEGs are involved in ion binding, cell development, metabolic processes, enriched in ECM–receptor interactions, steroid biosynthesis, together with aldosterone generation/discharge and Wnt/PI3K-Akt signaling pathways. In addition, we speculate alternate splice development events to have important role/s in regulating ovarian functional genomic expression profiles. These results provide essential knowledge aimed at scrutinizing pivotal biomarkers for yak ovarian activity, together with paving the way for enhancing researchers’ focus on improving yak reproductive performance.

## 1. Introduction

Yak (*Bos grunniens*) is an endemic cattle species centered on the Qinghai-Tibet Plateau in China and is distributed in the surrounding alpine pastoral areas [1]. Yak can provide local herdsmen with meat, milk, wool, labor, fuel, among other daily necessities [2]. It is an essential part of livelihoods and local herding culture/economy, known as the ‘boat of the plateau’. However, yak reproductive rates are relatively low compared to yellow cattle [3]. In addition, yaks have a delayed sexual maturity, and female yaks usually have the first estrus at the age of 1.5–2.5 years. Under the current production management level, yaks are generally calved every other year, with one in two years or two in three years [4]. Only 3.03% of yaks could estrus and breed during calving that year. Due to the influence of natural ecological conditions and forage supply of natural grassland, the reproductive performance of female yak showed obvious seasonal variations [5]. Xiang et.al [6] concluded that in the warm season (July to October) when pasture is abundant yak body condition and the nutritional level dictate endocrine regulation mechanism for the estrus season.

The ovary is the most important reproductive organ determining estrus activity and pregnancy of female animals. The histological structure of yak ovary is similar to that of cattle and buffalo. As early as 1999, Cui and Yu developed detailed comparisons and descriptions of yak ovary morphology and follicular system at differing ages [7] and found that the right ovary’s small follicular quantity was elevated in comparison to the left ovary [8]. Yu et al. [9] utilized light (LM) and transmission electron microscopy (TEM) techniques to observe several characteristic morphological variations in yak oocytes. Meng et al. [10] compared yak ovary structure in the estrus cycle, observing significant variations in cystic/atretic follicular quantity across differing estrus stages. The number and quality of follicles within ovaries are related to multiple factors, such as sex hormone levels, ovary volume, and size of the corpus luteum.

Ovarian development is a complex process that requires precise coordination of multiple molecules, cells, and tissues. At different stages of the reproductive cycle, there are numerous considerable variations in endocrine features and ovarian activity driving genomic/hormonal regulatory processes. Lan et al. [11] performed RNA-seq on adult yak ovary tissues, predicted and analyzed 6321 novel transcripts, and discovered several genes related to yak reproductive performance. Consequently, the group compared variations in ovarian transcriptome between yak and yellow cattle to further explore the specificity of yak reproduction [12]. Xu et al. [13] studied follicle growth/genomic expression profiling within yak ovarian tissue during hot/cold months, and found transcripts related to estrogen secretion and metabolic signaling pathways. However, there is currently no concomitant report on the comparison of yak ovarian activities with anestrus, estrus, and pregnancy.

This study aimed to deeply explore the basic physiological and molecular characteristics of yak ovaries, compare the internal differences of yak ovaries during anestrus, estrus, and pregnancy periods. Consequently, we compared the yak ovary follicular distribution at differing stages through H&E staining. Then, RNA-seq was employed for comparing transcriptomic data for all three stages and analyzing the differential genomic expression patterns. This study provides a reference for increased knowledge on yak reproduction specificity via investigating the molecular mechanism of yak ovarian physiological activity and furthering basic ovarian physiological research and the development/utilization of breed genetic resources.

## 2. Materials and Methods

### 2.1. Sample Collection

In this study, samples were collected from the ovaries of healthy adult yaks (6 years old), within one year and during three periods of anestrus (April), estrus (August), and pregnancy (December). Ten ovarian samples were collected in each period for observing and making tissue sections, and three samples in each period were used for RNA-seq. All samples were collected from yaks that were euthanized within one farm, with such samples collected within 60 min post-mortem. A section of the ovarian sample was stored within 10% neutral-formaldehyde fixative, with the remaining sample segment flash-frozen within liquid nitrogen. All samples were obtained through Menyuan Hui Autonomous County, Haibei Tibetan Autonomous Prefecture, Qinghai Province, China (37°39″ N, 101°62″ E). All animal handling/experimental protocols were conducted in line with regulations set by the China Council on Animal Care and the Ministry of Agriculture of the People’s Republic of China. The Animal Care and Use Committee of the Lanzhou Institute of Husbandry and Pharmaceutical Sciences Chinese Academy of Agricultural Sciences accepted all animal handling protocols (Permit No: SYXK-2014-0002).

### 2.2. H&E Staining

The fixed ovarian tissues were dehydrated using a fully-automatic dehydrator [Junjie Electronics Co., Ltd.™, Wuhan, China] (alcohol dehydration time: 75%—4 h, 85%—2 h, 95%—1 h, 100%—0.5 h, 100%—0.5 h, 100%—0.5 h, 100%—0.5 h, xylene—10 min, xylene—10 min, paraffin—1 h, paraffin—2 h, paraffin–3 h), embedded and sliced. Then, the procedure consisted of: deparaffinizing sections to water (xylene I for 20 min, xylene II for 20 min, absolute ethanol I for 5 min, absolute ethanol II for 5 min, 75% alcohol for 5 min, and tap water rinsing); hematoxylin staining (10 min); water-wash step (60 s); differentiation with hydrochloric acid and alcohol (5 s); water-wash step (60 s); transfer into warm water at 50 °C or weak alkali the aqueous solution turns into blue coloration; water-wash step (60 s); placed into 85% alcohol for 180 s; eosin staining (180 s); water-wash step (3 s); gradient-alcohol dehydrating steps; xylene transparency step; gum-based neutral sealing. All samples were subjected to standardized operating protocols (SOPs) applied for pathology laboratory assessment processes, and consequently underwent microscopic examination and imaged by Panoramic 250^®^ digital slice scanner [Tangier Electronics Co., Ltd.™, Jinan, China].

### 2.3. RNA Sequencing

Total RNA was collected through TRIzol^®^ [Invitrogen™, Carlsbad, CA, USA] according to manufacturer protocol., followed by RNA quantitation/purity assessment through NanoDrop ND-1000^®^ [NanoDrop™, Wilmington, DE, USA]. RNA integrity was evaluated through Bioanalyzer 2100^®^ [Agilent™, Palo Alto, CA, USA] and verified by agarose electrophoresis. Samples having a concentration of >50 ng/μL, RIN value > 7.0, OD260/280 > 1.8, total RNA >50 μg were deemed to meet the requirements for downstream studies. Through oligo(dT)-based magnetic beads [Dynabeads Oligo (dT)^®^, catalog number 25-61005, Thermo Fisher™, Waltham, MA, USA], mRNA containing PolyA (polyadenylic acid) were specifically captured through two purification rounds. All captured mRNA was fragmented using the NEBNext^®^ Magnesium RNA Fragmentation Module [Catalog No. E6150S, New England BioLabs (NEB)™, Ipswich, MA, USA] with high-temperature conditions, at 86 °C for seven minutes. Consequently, cDNA was created through SuperScript II Reverse Transcriptase^®^ [Invitrogen™, cat. 1896649, Carlsbad, CA, USA], required for producing U-labeled second-stranded DNA employing *E. coli* DNA polymerase I [NEB™, cat.m0209, Ipswich, MA, USA], RNase H [NEB™, cat.m0297, Ipswich, MA, USA] and dUTP Solution [Thermo Fisher™, cat.R0133, Waltham, MA, USA]. A-base was subsequently introduced onto blunt ends for each strand, prior to ligation onto indexed adapters. Individual adapters contained a T-base overhang to enable the ligation step. Single/dual-index adapters were ligated onto DNA fragments, while size selection was conducted through AMPureXP^®^ beads. Following heat-labile UDG enzyme [NEB™, cat.m0280, Ipswich, MA, USA] activity on U-labeled second-stranded DNA, all ligated products underwent PCR under such optimizations: single-step denaturation at 95 ℃ for three minutes; eight denaturing cycles at 98 ℃ for 15 s, annealing at 60 ℃ for 15 s, and extension at 72 ℃ for 30 s; single-step extension at 72 ℃ for five minutes. The mean insert size for finalized cDNA library was 300 ± 50 bp. Consequently, 2 × 150 bp paired-end sequencing (PE150) was conducted on an illumina™ Novaseq 6000^®^ [LC-Bio Technology CO., Ltd.™, Hangzhou, China] according to the manufacturer’s protocol.

### 2.4. Sequencing Analysis

The ‘off-machine’ raw data format was fastq. Fastp software was employed to perform quality control on sample raw data, including removal of joints, duplicates, together with reduced-quality sequences in order to attain CleanData format for fastq [14]. HISAT2 was consequently employed to align the obtained CleanData onto the designated genome (Latin: bos grunniens, genome version-101), and consequently, obtain bam file format [15]. Gene assembly and quantification software was StringTie [16], while the quantification method was Fragments per Kilobase Million (FPKM)(total exon fragments/mapped reads (millions) × exon length (kB)), with edgeR employed for variation analyses [17]. The GO analysis and KEGG analysis were separately completed through online websites G: profiler (https://biit.cs.ut.ee/gprofiler/gost, accessed on 20 August 2021) and KOBAS 3.0 (http://kobas.cbi.pku.edu.cn, accessed on 16 September 2021).

### 2.5. Real-Time Quantitative PCR (RT-qPCR)

Reverse transcription of RNA was performed through Evo M-MLV RT Kit^®^ with gDNA Clean for qPCR ‖ [AG™, Hunan, China]. Primer set designs for investigated genes were developed through the NCBI website, and the sequences are shown in Appendix A. Consequently, SYBR Green master mix [Yeasen™, Shanghai, China] and the LightCycler^®^ System [CFX96TM Optics Module, Singapore] were both employed to perform RT-qPCR. The reaction volume was 20 µL, including 2x Precision PLUS^®^ Master Mix 10 µL, cDNA (25 ng) 1 µL, primers (forward/reverse) 1 µL each (300 nmol), together with 7 µL RNase/Dnase-free water. PCR running conditions were: 95 °C for three minutes, followed by 39 cycles of 95 °C for 10 s/55 °C for 30 s. The data were represented as mean ± SE (n = 3). Glyceraldehyde-3-phosphate dehydrogenase (GAPDH) served as the reference gene for normalization, while the 2^−ΔΔCT^ calculation method unified transcriptomic expression. All statistical assessments were performed by SAS 9.4^®^ statistical software, with *p* < 0.05 deemed to confer statistical significance.

## 3. Results

### 3.1. Ovarian Surface Observation

The adult yak’s ovary is slightly flattened and oval, with raised follicles and corpus luteum visible on the surface (Figure 1). Compared to other cattle, yak has reduced ovaries, shorter ovarian mesentery, and relatively fixed positions, though overall structure is similar. Individual yak ovary pairs of each yak occurred in differing states. The overall length of the larger side is approximately 20–26 mm. Typically, there is a large corpus luteum, which almost occupies the whole ovary space. Through manual evaluation of yak ovary dimensions (greater region) during differing time periods, it was found that the mean yak ovary diameter during pregnancy was slightly larger, at 24.7 mm, followed by the estrus period, and with the most reduced diameter occurring in the anestrus period (Table 1). By observing the ovarian surface, it could be found that there was a corpus rubrum with a diameter of approximately 12 mm in most of the ovaries during estrus, while there was a corpus luteum with a diameter of approximately 20 mm in the ovary during pregnancy, corpus luteum can also be seen in individual ovaries during estrus. The size and duration of the corpus luteum depended on the mother’s state, such as fertilization or pregnancy.

There are follicles of differing sizes in the ovary, typically one with a diameter of >5 mm, and several small follicles of 2–5 mm. The number of follicles between 2–3 mm within ovaries during the anestrus period have the highest follicular population density, with additional immature follicles, while the number of small follicles is reduced, due to the occupation of the corpus luteum/corpus rubrum during the estrus and pregnancy periods.

### 3.2. Histological Characteristics of Follicles

According to the description of histological characteristics for varying follicles by Yu et al. [9], comparison of ovarian follicles at three differing stages of anestrus, estrus, and pregnancy, the characteristics of different follicles are shown in Figure 2. It was found that there were increased numbers of follicles at all levels within ovarian tissue during estrus, with additional developing/mature follicles being observed. The number of follicles during anestrus and pregnancy were similar, with fewer developing follicles, and increased numbers of atretic follicles. Figure 3 shows the microscopic anatomical structure of immature Graafian follicles of the yak ovary, from which it could be seen that the granular cell layer was arranged disorderly.

### 3.3. Raw Sequencing Data and Descriptive Statistics

In order to attain transcriptomic mapping for yak ovaries, such tissue was assessed through RNA-seq during three differing periods, namely, anestrus (YO-A), estrus (YO-F), and pregnancy (YO-P). Readings with connector, low mass readings, together with low mass bases were excluded through Trimomatic v0.36 software. A total of 72.35 G of raw bases were obtained in this sequencing, and each sample averaged 8.04 G of raw bases (Appendix A). Following fastp to filter out unqualified sequences, a total of 65.16 G in valid bases were obtained, accounting for 90%. The principal component analysis (PCA) results of nine samples in three groups showed that YO-F2 deviated from other samples (Figure 4a), so this sample was excluded from future analyses. A total of 19,738 transcripts were identified in the three groups, including 18,439, 17,758, and 18,260 transcripts within YO-A, YO-F, and YO-P groups, accordingly. Box plot and density graphs of gene expression for each sample are shown in Figure 4b,c. Approximately 25% of all investigated genes had reduced expression profiles (FPKM < 1). Approximately 72% of such genes had an expression level of FPKM 1–100, with just 2% having gene expression profiles of FPKM 100–1000. In essence, a minority of investigated genes were up-regulated (FPKM > 1000), with the maximum FPKM value reaching 8906.

### 3.4. Differentially Expressed Gene Identification

The three groups (YO-A, YO-F, and YO-P) were segregated into three comparative groups: YO-FvsA, YO-FvsP, and YO-PvsA. According to difference multiple | logfc | ≥ 1 and *p* < 0.05 as the standard, the screened genes were differentially expressed. This study screened 294, 410, and 273 differentially expressed genes (DEGs) across all comparative groups (Figure 4d). DEGs included 161 up-regulated genes/133 down-regulated genes within YO-FvsA (Appendix A), 208 up-regulated genes/202 down-regulated genes within YO-FvsP (Appendix A), together with 125 up-regulated genes/148 down-regulated genes within the YO-PvsA group (Appendix A), respectively. The study selected the top-ranking 100 DEGs for cluster analysis and displayed genomic expression profiles within differing samples through heat maps (Figure 4e).

### 3.5. Gene Ontology Analysis of Differentially Expressed Genes

There are three ontologies in GO (Gene Ontology) describing genomic molecular roles, any cellular constituents, and physiological activities involved. This study conducted GO analysis on DEGs (*p* < 0.05), across all three comparison groups, with specific results shown in the Appendix A.

Regarding molecular function category, DEGs within the YO-FvsA group were significantly enriched in binding, calcium ion binding, and cargo receptor activity (Figure 5a). DEGs in the YO-FvsP group were significantly enriched in signaling receptor binding, receptor–ligand activity, calcium ion binding, and G protein-coupled receptor binding (Figure 5b). DEGs within the YO-PvsA group were highly enriched for signaling receptor binding, catalytic function, together with protein–lipid complex binding (Figure 5c). Regarding the cellular components category, the DEGs of all three groups were mainly enriched in the plasma membrane, cytoplasm, cell surface, and cilium. Regarding the biological process category, the DEGs of the three groups were all significantly related to cell-based growth processes, modulating developmental processes, cellular population proliferation, and differentiation, and metabolic processes. In addition, this study revealed that each comparison group enriched to unique processes, such as the DEGs in the YO-FvsP group, that were significantly enriched in enzyme-linked receptor/protein signaling pathway and enhancing steroid metabolic processes, while DEGs in the YO-PvsA group were significantly enriched in reproductive process and reproduction.

### 3.6. KEGG Pathway Analysis of Differentially Expressed Genes

Consequently, KEGG (Kyoto Encyclopedia of Genes and Genomes) analysis on DEGs for all three comparison groups was conducted in Appendix A. Regarding the YO-FvsA group, 163 genes were highly enriched for 28 pathways (*p* < 0.05), which were mainly related to ECM–receptor interaction, PI3K-Akt signaling pathway, parathyroid hormone generation, discharge, and activity (Figure 5d). A total of 237 genes were highly enriched within 21 pathways, such as ECM–receptor interaction, ovarian steroidogenesis, PPAR, and Wnt signaling pathways, within the YO-FvsP group (Figure 5e). Regarding the YO-PvsA group, 168 genes were highly related to 38 pathways, such as the Wnt signaling pathway, aldosterone synthesis and secretion, cAMP signaling pathway, steroid biosynthesis, PPAR signaling pathway, ECM–receptor interaction, IL-17 signaling pathway, among others (Figure 5f).

### 3.7. RT-qPCR Validation

In order to validate RNA-seq results, *SULT1C3*, *HSPA6*, *ADGRB3*, *ECM1*, *ITGAD*, *STAR*, and *IRF4* were selected for RT-qPCR analyses. According to the results of such evaluations, the expression profile in such DEGs varied in at least one of the three control groups. *HSPA6* and *ECM1* were differentially expressed in all groups; *ITGAD* was up-regulated at anestrus; *STAR* was up-regulated at pregnancy. As shown in Figure 6, the expression profiles for such DEGs conformed to the RNA-seq results.

### 3.8. Alternative Splicing Analysis

Alternative splicing (AS) describes the events in which RNA exons, produced mainly by the transcription of genes or mRNA precursors, are reconnected post-RNA splicing, in multiple manners [18]—it is common in eukaryotic genes. This study employed ASprofile software to perform the qualitative analysis and statistics for alternative splicing events, within individual samples, on a genetic model predicted by Stringtie (transcript.gtf). The study analyzed alternative splicing events, according to the annotation data for yak gene structure. A total of seven main types of alternative splicing events were detected as follows: Exon skipping (SKIP) and cassette exons (MSKIP), retention of single (IR) and multiple (MIR) introns, alternative exon ends (AE), alternative transcription start site (TSS), and alternative transcription termination site (TTS). The remaining five types of fuzzy-boundary alternative splicing were as follows: approximate SKIP (XSKIP), approximate MSKIP (XMKIP), approximate IR (XIR), approximate MIR (XMIR), and approximate AE (XAE), are shown in Figure 7. TSS and TTS types accounted for approximately 40%, respectively.

## 4. Discussion

In recent years, due to the advantages of low cost and rapid transcriptome sequencing technology, researchers used RNA-seq technology to study the differential expression of mRNAs [19,20]. The reproduction process in mammals has received considerable attention. In order to explore in great detail the reproductive mechanism in yaks, this study compared the number of follicles within ovaries during anestrus, estrus, and pregnancy through histological methods, together with analysis and identifying the entire mRNA sequences expressed in yak ovaries during differing reproductive stages through transcriptome sequencing technology, and found considerable volumes of useful information regarding yak reproduction physiology.

The ovary is a dynamic organ whose size and activity are constantly changing throughout life [21]. During ovulation, ovarian volumes increase slightly. Our results indicate that the ovaries are at their largest size during pregnancy, which may be related to the increase in female hormones during pregnancy. The most important activity within the ovary is follicular development. In recent years, more attention has been paid to the transition between different stages of follicular development, maturation, atresia or cell differentiation, and the regulation of ovulation processes. Prior to sexual maturity, most primordial follicles can only develop to the stage of antral follicles, and atresia will eventually occur. Following sexual maturity, a few follicles are rescued by gonadotropins and continue to grow. During the estrus season, post-follicle selection and domination, one or several follicles mature and ovulate, with >99% of all other follicles that enter the growth group being atresic. During pregnancy, a period of non-ovulatory follicular presence persists on the ovary, though this follicular population density is severely reduced in comparison to the non-pregnant state, with the existing follicular variations being similar to those in the pre-estrus period. The follicular pool within yak ovaries is much smaller than for other cattle species (such as cattle and dairy cows), and the follicular atresia rate is relatively high. Similarly, in our tissue section results, we found that there are increased numbers of follicles at all levels within ovarian tissue during estrus, while the number of follicles within ovarian tissue during anestrus and pregnancy is significantly reduced, with more atretic follicles.

Follicle development is complex, involving the transmission of autocrine and paracrine signals in the ovary, as well as the exchange of endocrine signals between the ovary and pituitary gland. Studies have shown that activation and development of preantral follicles do not depend on gonadotropins, though are mainly regulated by ovarian factors [22]. With continuous follicular growth, the later stage of development is mainly regulated by endocrine mechanisms, especially gonadotropins FSH and LH [23,24]. Our present study found that DEGs are related to follicular development and hormone metabolism. Genes related to degeneration of atretic follicles [25] were found across all three comparison groups, including *COL1A2*, *COL2A1*, *COL4A6*. Genes such as *NR4A1* and *THBS2* were linked to luteal lysis and apoptosis [26,27] and were found in the comparison groups of anestrus and estrus. Many genes related to reproduction were also found in the comparison group between estrus and pregnancy, such as *NPR1*, related to oocyte meiosis recovery [28] and ovulation [29]; *PTGS2*, involved in luteal formation [30]; *SCARB1* and *STAR*, maintaining steroid synthesis [31,32,33,34]; *BMP6* and *BMP15*, regulating hormone secretion [35], granulosa cell differentiation and follicular development [36,37]. Genes such as *WNT2B*, *WNT2*, *WNT11* related to embryonic development [38,39] and maintenance of maternal pregnancy [40,41] were found in the comparison groups for pregnancy and anestrus.

Therefore, we performed a comparative analysis of the three groups and performed an enrichment analysis of the differential DEGs identified between any two groups. GO/KEGG evaluations highlighted that many DEGs are implicated within follicle growth/ovary-based steroid generation. According to the GO analysis results of the three comparison groups, “binding” accounts for the largest proportion within the molecular function category, which has been reported and confirmed previously [42], indicating that this role could have pivotal parts within typical yak ovarian physiology. Metal ion metabolism has pivotal parts within follicular growth. Case in point, Ca^2+^ exerts pivotal positions for modulating oocyte meiotic development, together with initial embryonic progression [43]. In our results, it was confirmed that ion, metal ion, and calcium ion binding are essential molecular activities occurring throughout follicle growth. DEGs become significantly enriched within biological processes such as multicellular whole organisms, cellular development, and regulation of developmental processes. This study speculates such processes to possibly be implicated within ovarian granulosa cell proliferation and the growth and development of follicles. Cell composition results show that cytoplasm, extracellular matrix/organelles are predominant cell-based constituents for follicular growth.

We found an interesting result through KEGG analysis, that the DEGs for all three comparison groups were significantly enriched in the ECM–receptor interaction pathway. The extracellular matrix regulates cell proliferation, differentiation, and survival by interacting with its surface receptors (ECM receptors), thereby affecting oocyte maturation and follicular growth. Chu et al. [44] found that ECM receptor interaction plays an important role in maintaining normal folliculogenesis and estrus, by comparing sows in diestrus and estrus. Studies have reported that ECM regulates cumulus development [45], granulosa cell growth and adhesion [46], and steroid hormone synthesis [47]. Steroid hormones, including cortisol, aldosterone, and estradiol are synthesized by cholesterol within specialized endocrine ovarian cells and systemically discharged when required [48]. The expression of steroid sex hormones affects a series of primary and secondary sexual characteristics for follicles and regulates the process of oogenesis, follicle generation, ovulation, and pregnancy [49,50]. Wnt signaling also modulates multiple ovarian activities, such as follicle growth, granulosa cell (GC) proliferative and differentiating properties, steroid production, and ovulating function [51]. DEGs in the two comparison groups of pregnancy and estrus/pregnancy and anestrus were significantly enriched in steroid biosynthesis, aldosterone synthesis/secretion, and the Wnt signaling pathway, indicating that these three pathways are involved in ovarian follicle growth, ovulation, pregnancy, and other physiological activities. Closely resembling the results from this investigation, Chen et al. [52] compared dysregulated mRNA expression profiles of high-fertility and low-fertility sheep ovaries and found that several such genes are implicated within steroid biosynthesis, TGFβ, Wnt, Notch signaling pathway.

In addition, many studies have shown that activation of the renin-angiotensin-aldosterone system (RAAS) is necessary for maternal adaptation during pregnancy [53,54]. The PI3K/Akt signaling pathway is pivotal for animal germ cell development, with an emerging body of evidence found within animal reproduction. We found a significantly enriched PI3K-Akt signaling pathway within the comparison group concerning estrus and anestrus. Recent emerging study outcomes confirmed that the PI3K/Akt signaling pathway and selected downstream effector molecular dysregulations occurred during follicle formation, growth, ovulation, and luteinization [55,56]. This suggested that PI3K/Akt signaling pathway has pivotal parts within ovarian follicular growth. There are a large number of atresia follicles on the ovary during the anestrus period, and granulosa cell apoptosis has pivotal parts in follicular atresia. During estrus, PI3K stimulation can promote Akt phosphorylation and promote follicular survival/growth. Follicular recruitment is regulated by granulosa cell proliferation, Notch, and PI3K/Akt pathway interaction [57]. Therefore, we speculate that the PI3K/Akt signaling pathway can regulate follicular atresia, recruitment, and growth in ovaries.

Alternative splicing (AS) is recognized as an essential method for regulating genomic expression, indirectly orchestrating physiological activities in higher organisms. TSS is the predominant AS event, followed by TTS and SKIP. Ovarian activity represents non-simplistic physiological processes involving transcriptional regulation for considerable numbers of genes [58]. Studies have shown that AS luteinizing hormone receptors within sheep ovaries during estrus are expressed in differing cellular development stages [59]. AS variants of FSHR could be involved in follicular dynamics during the follicular wave of the estrous cycle in sheep [60]. Qiu et al. [61] found that the vascular endothelial growth factor-A (VEGFA) protein family regulates the progression of follicles and luteinization in mammalian ovaries through alternative splicing. Multiple genes implicated within gonad growth/hormonal metabolic processes have AS, suggesting that AS has pivotal parts within ovarian functional gene expression modulation [62,63].

## 5. Conclusions

Our present study compared the number of ovarian follicles during anestrus, estrus, and pregnancy through histological methods, and found that ovaries during differing reproductive periods had specific variations in tissue morphology. A total of 18,439, 17,758, and 18,260 transcripts were obtained in YO-A, YO-F, and YO-P through RNA-seq technology, and screened 294, 273, and 410 DEGs across three comparative groups YO-FvsA, YO-FvsP, and YO-PvsA, respectively. Our research found that DEGs are associated with follicle growth/hormonal metabolic processes. GO/KEGG evaluations revealed such DEGs to be implicated in follicle growth and ovary-based steroid generation, including ion binding, multi-cellular whole-organism-level processes, cell-level developmental processes, together with modulating developmental processes. In addition, they were enriched in ECM–receptor interactions, steroid/aldosterone generation and discharge, and Wnt signaling/PI3K-Akt signaling pathways. Consequently, by analyzing the transcriptome of yak ovary in anestrus, estrus, and pregnancy stages, this paper explores the functions of specific mRNA in the ovary at different stages in the reproductive physiology, such as hormone discharge and follicular growth, to lay the foundation for improving the reproductive performance of yak.

## Figures and Tables

**Figure 1 biology-10-01229-f001:**
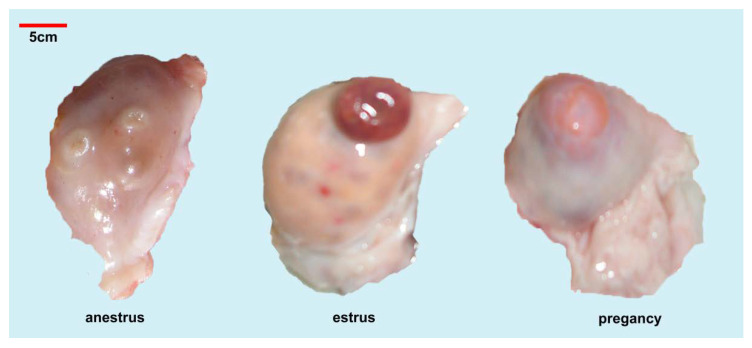
The ovarian morphology of yak in different states.

**Figure 2 biology-10-01229-f002:**
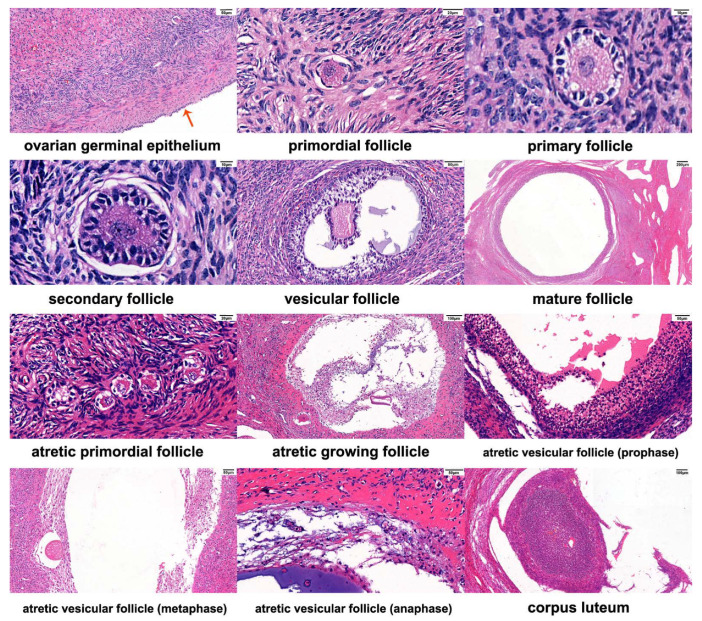
Diagram of the internal follicular structure within yak ovary.

**Figure 3 biology-10-01229-f003:**
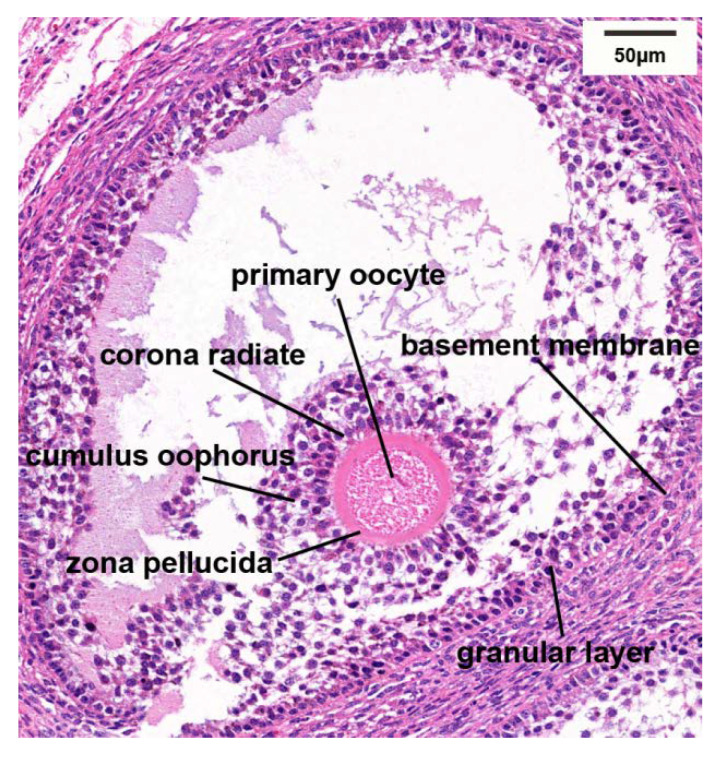
Microanatomic structure of immature Graafian follicle in yak ovary.

**Figure 4 biology-10-01229-f004:**
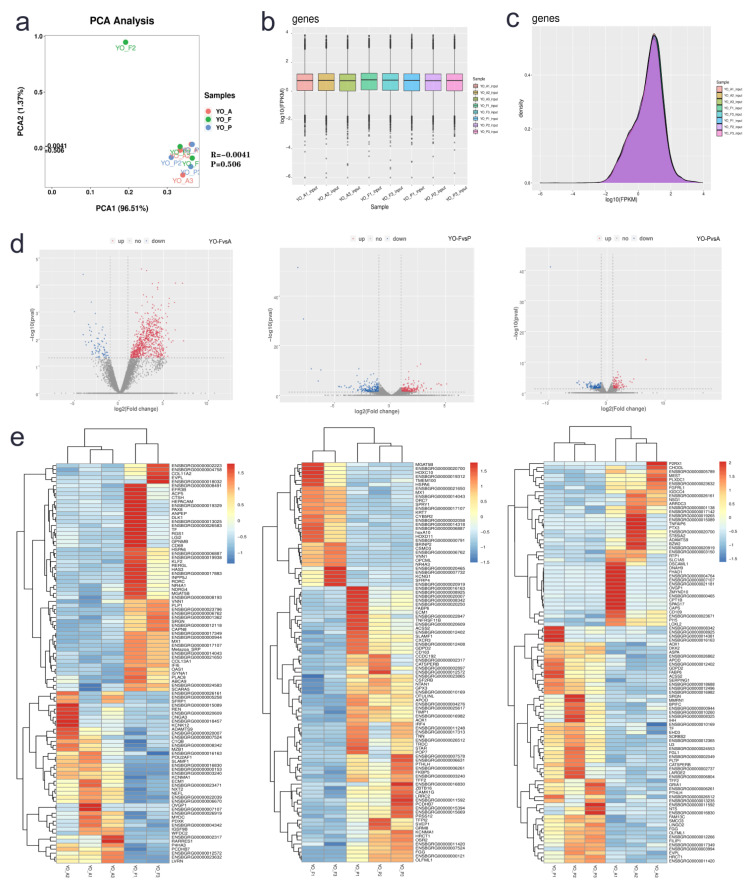
RNA-seq/DEG identification results for yak ovary. (**a**) PCA results of nine yak ovary samples. (**b**) Box plot graph for individual sample genomic expression. (**c**) Density graph for individual sample genomic expression. (**d**) DEG volcano map across three control groups; YO-FvsA, YO-FvsP, and YO-PvsA. (**e**) Cluster analyses results for top-ranking 100 DEGs within YO-FvsA, YO-FvsP, and YO-PvsA groups. In order to intuitively reflect the cluster expression pattern in an enhanced manner, log10 (FPKM + 1) was employed for genomic expression visualization on non-biological repetitions, while for biological repetitions, the differential gene FPKM was employed to display gene expression through Z-values (Zsamplei = (log2(Signal samplei)-Mean (log2(Signal) of all samples)/(Standard deviation (log2(Signal) of all samples)).

**Figure 5 biology-10-01229-f005:**
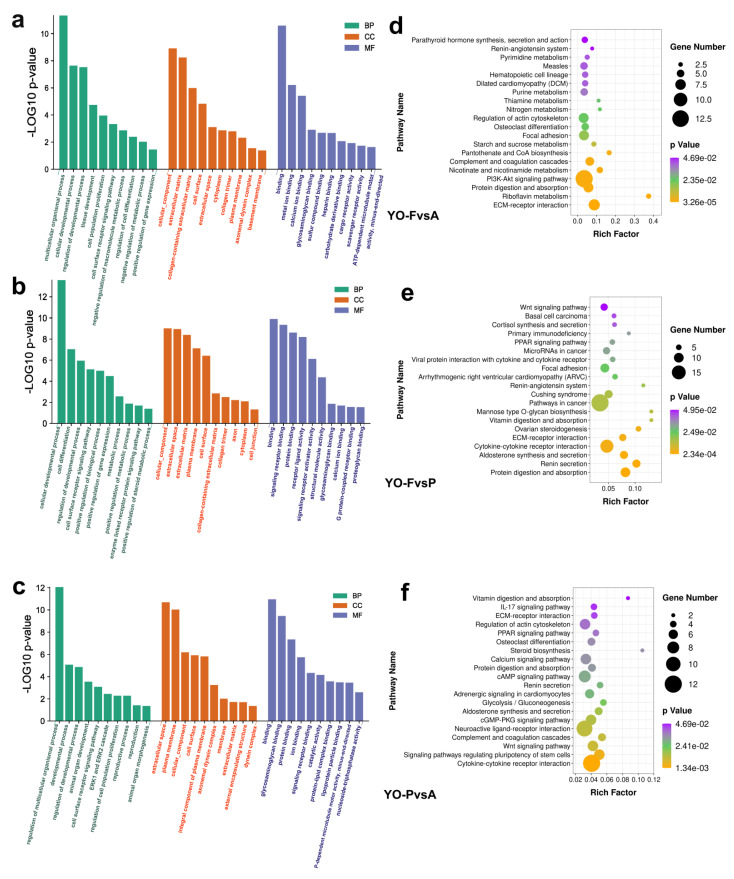
GO/KEGG analyses for DEGs across YO-FvsA, YO-FvsP and YO-PvsA groups. (**a**) GO analyses for DEGs within YO-FvsA groups. (**b**) GO analyses for DEGs within YO-FvsP groups. (**c**) GO analyses for DEGs within YO-PvsA groups. (**d**) KEGG analyses for DEGs within YO-FvsA groups. (**e**) KEGG analyses for DEGs within YO-FvsP groups. (**f**) KEGG analyses for DEGs within YO-PvsA groups.

**Figure 6 biology-10-01229-f006:**
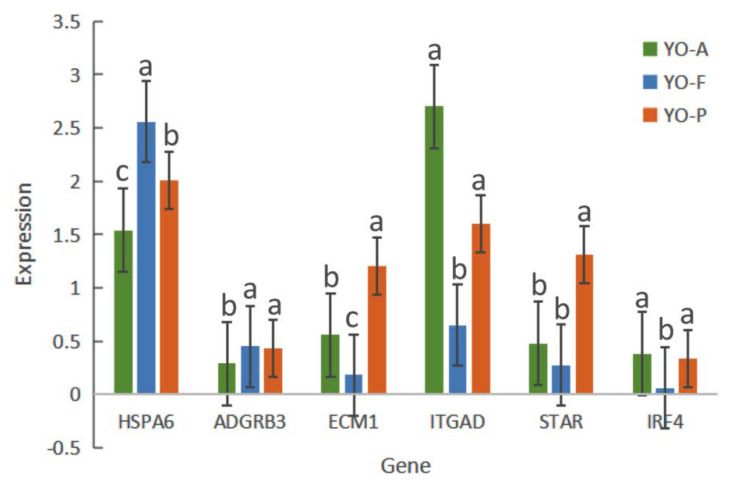
RT-qPCR results for *SULT1C3*, *HSPA6*, *ADGRB3*, *ECM1*, *ITGAD*, *STAR*, and *IRF4* within YO-A, YO-F, and YO-P groups. Different letters between the two groups represent significant differences between the two groups.

**Figure 7 biology-10-01229-f007:**
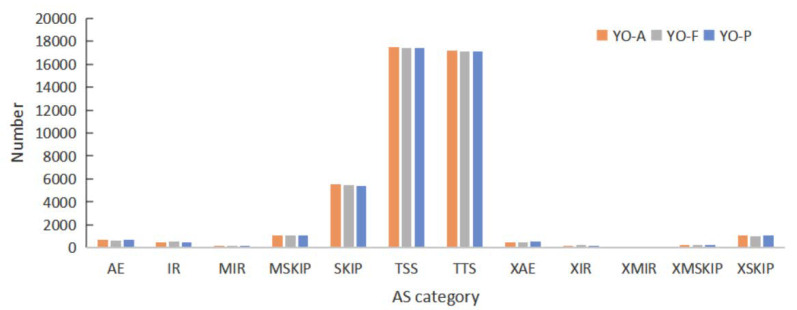
The number of differing AS events within YO-A, YO-F, and YO-P groups.

**Table 1 biology-10-01229-t001:** Statistics of ovarian size and number of follicles during differing periods.

Period	Size	Corpus Luteum or Corpus Rubrum	Number of Differing Follicles		
			2–3 mm	4–5 mm	>5 mm
Anestrus	22.5 mm	none	9	1.5	1
Estrus	23.5 mm	corpus rubrum/corpus luteum	5	2	1
Pregnancy	24.7 mm	corpus luteum	4.7	3.5	0.7

## Data Availability

Raw reads of transcriptome sequencing of ruminal epithelium are available at GEO. https://www.ncbi.nlm.nih.gov/geo/query/acc.cgi?acc=GSE180401.

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
