# Peer review of "Changes in Transcriptomic Profiles in Different Reproductive Periods in Yaks"

_biology, 2021, doi:10.3390/biology10121229_

Round 1

Reviewer 1 Report

General comments

In this manuscript, the authors reported their investigation of transcriptomic regulation of ovarian activity over different reproductive periods in Yaks. The study was reasonably designed and the results are interesting and provided useful information about the ovarian gene expression profiles during different reproductive periods in this animal species.

Special comments

  1. The title. I suggest changing the title to “Changes in Transcriptomic Profiles in Different Reproductive Periods in Yaks” as the authors compared the physiological changes in ovarian activity between different periods and they cannot be regarded as “Dysregulation”.
  2. Line 13. Change “this study detect the overall gene expression in differing reproductive stages” to “This study investigated the global gene expression profiles in different reproductive stages”.
  3. Line 20. Change “ovary” to “ovarian activities”.
  4. Line 22. Change “differing” to “different”.
  5. Line 31. Change “for regulating” to “in regulating”.
  6. Line 33. Change “researcher” to “ researchers’ “.
  7. Line 43. Change “yak sexual maturity is delayed” to “yaks have a delayed sexual maturity”.
  8. Line 73. Change “ovary” to “ovarian activities”.
  9. Line 77. Change “compare” to “compared”.
  10. Line 81. Change “specificity exploring” to “specificity via investigating”.
  11. Line 114. Section 2.3. Please give the depth of the RNA-Seq.
  12. Line 142. Change “Credit analysis” to “Sequencing analysis”.
  13. Line 114. Please give the full name of FPKM. Also please keep all abbreviations consistent throughout the manuscript. Give the full name at first mention and then use the abbreviation throughout the manuscript.
  14. Line 156. Change “template is” to “sequences are”
  15. Lines 63 - 64. Change “a normalization gene” to “the reference gene for normalization”.
  16. Line 179. Change “the most ovary” to “most of the ovaries”.
  17. Line 204. Change “dentificated genes” to “descriptive”.
  18. Line 221. Delete “selection”.
  19. Section 3.4. Did you use any false discovery rate control for multiple tests when identifying the DEGs?
  20. Figure 4. Words in Figure 4a are difficult to read.
  21. Line 317. Change “The ovary” to “Ovary”.
  22. Line 343. Change “research” to “present study”.
  23. Lines 355 - 356. Rephrase.
  24. Line 360. Change “confirmed” to “confirmed previously”.
  25. Line 423. Change “In essence, this” to “our present”.
  26. Lines 435 - 439. Rephrase.

Author Response

  1. The title. I suggest changing the title to “Changes in Transcriptomic Profiles in Different Reproductive Periods in Yaks” as the authors compared the physiological changes in ovarian activity between different periods and they cannot be regarded as “Dysregulation”.

Response 1:Thank you for your advice, we have revised the title according to your suggestion.

  1. Line 13. Change “this study detect the overall gene expression in differing reproductive stages” to “This study investigated the global gene expression profiles in different reproductive stages”.

Response 2:Thank you for your advice, we have revised according to your suggestion.

  1. Line 20. Change “ovary” to “ovarian activities”.

Response 3:Thank you for your advice, we have revised according to your suggestion.

  1. Line 22. Change “differing” to “different”.

Response 4:Thank you for your advice, we have revised according to your suggestion.

  1. Line 31. Change “for regulating” to “in regulating”.

Response 5:Thank you for your advice, we have revised according to your suggestion.

  1. Line 33. Change “researcher” to “ researchers’ “.

Response 6:Thank you for your advice, we have revised according to your suggestion.

  1. Line 43. Change “yak sexual maturity is delayed” to “yaks have a delayed sexual maturity”.

Response 7:Thank you for your advice, we have revised according to your suggestion.

  1. Line 73. Change “ovary” to “ovarian activities”.

Response 8:Thank you for your advice, we have revised according to your suggestion.

  1. Line 77. Change “compare” to “compared”.

Response 9:Thank you for your advice, we have revised according to your suggestion.

  1. Line 81. Change “specificity exploring” to “specificity via investigating”.

Response 10:Thank you for your advice, we have revised according to your suggestion.

  1. Line 114. Section 2.3. Please give the depth of the RNA-Seq.

Response 11:Thank you for your comments, due to the spatio-temporal and temporal specificity of RNA expression, the number of RNA expression in different tissues at different times or at the same time is quite different. Therefore, the amount of RNA-Seq data generally does not need to be expressed by the sequencing depth. The reference genome size of yak is 2.7G, RNA-Seq adopts paired-end sequencing technology, each cluster is tested once from both ends, and each test is 150bp, and finally each yak ovary sample obtained approximately 8G of raw bases.

  1. Line 142. Change “Credit analysis” to “Sequencing analysis”.

Response 12:Thank you for your advice, we have revised according to your suggestion.

  1. Line 114. Please give the full name of FPKM. Also please keep all abbreviations consistent throughout the manuscript. Give the full name at first mention and then use the abbreviation throughout the manuscript.

Response 13:Thank you for your comments, I’m sorry I didn’t find FPKM in line 114. I found FPKM in line 149, and added its full name Fragments per Kilobase Million, and the abbreviation used in subsequent FPKM.

  1. Line 156. Change “template is” to “sequences are”

Response 14:Thank you for your advice, we have revised according to your suggestion.

  1. Lines 63 - 64. Change “a normalization gene” to “the reference gene for normalization”.

Response 15:Thank you for your advice, we have revised according to your suggestion.

  1. Line 179. Change “the most ovary” to “most of the ovaries”.

Response 16:Thank you for your advice, we have revised according to your suggestion.

  1. Line 204. Change “dentificated genes” to “descriptive”.

Response 17:Thank you for your advice, we have revised according to your suggestion.

  1. Line 221. Delete “selection”.

Response 18:Thank you for your advice, we have revised according to your suggestion.

  1. Section 3.4. Did you use any false discovery rate control for multiple tests when identifying the DEGs?

Response 19:Thank you for your comments, in this article, we used general methods to screen DEGs, and the screening threshold was dff.p<0.05 and |diff.log2.fc|≥1. Considering that we want to keep some interesting genes, our data did not use stricter filtering methods, such as Q-value or FDR value.

  1. Figure 4. Words in Figure 4a are difficult to read.

Response 20:Thank you for your comments, we have reformatted Figure 4 and adjusted the font size and clarity.

  1. Line 317. Change “The ovary” to “Ovary”.

Response 21:Thank you for your advice, we have revised according to your suggestion.

  1. Line 343. Change “research” to “present study”.

Response 22:Thank you for your advice, we have revised according to your suggestion.

  1. Lines 355 - 356. Rephrase.

Response 23:Thank you for your comments, we have changed “Consequently, enrichment analyses were conducted on DEGs identified across three comparison groups.” to “Therefore, we performed a comparative analysis of the three groups and performed an enrichment analysis of the differential DEGs identified between any two groups.”

  1. Line 360. Change “confirmed” to “confirmed previously”.

Response 24:Thank you for your advice, we have revised according to your suggestion.

  1. Line 423. Change “In essence, this” to “our present”.

Response 25:Thank you for your advice, we have revised according to your suggestion.

  1. Lines 435 - 439. Rephrase.

Response 26:Thank you for your comments, we have changed “Consequently, the authors believe that such data will support researchers to further recognize the value of mRNAs within reproduction physiology, such as hormonal discharge and follicle growth, as well as detailed investigations upon putative genes, and contributing to the advancement of yak functional genomic resources and future breeding practices / reproductive regulation research.” to “Consequently, by analyzing the transcriptome of yak ovary in anestrus, estrus and pregnancy stages, this paper explores the functions of specific mRNA in ovary at different stages in the reproductive physiology, such as hormone discharge and follicular growth, to lay the foundation for improving the reproductive performance of yak.”

Reviewer 2 Report

Comments on the manuscript:

“Transcriptomic Dysregulation Analysis of Yak Ovary in Differing Reproductive Periods”

The reproductive characteristics of the yak are known but the molecular regulation mechanism of the ovarian activity of the yak remains to be explored. The aim of this study was to examine the physiological and molecular characteristics of the yak ovaries, depending on the different periods of the reproductive cycle. To do this, the authors studied the follicular distribution at different stages thanks to staining with hemalun-eosin, the RNA-sequencing.

This article brings elements to the knowledge of the regulation of the ovarian cycle of the yak and deserves to be published after, however, some improvements. Here are some remarks.

Line 38: use italics to write “Bos grunniens”.

Line 77: write “periods” instead of “peroids”.

Line 86: the authors write: “In this study, samples were collected from the ovaries of healthy adult yaks (6 years old), within one year and during three periods of anestrus (April), estrus (August) and pregnancy (December),…” Specify how many individuals were studied in each period.

Line 104: the authors write “Then the procedure consisted of: deparaffinizing sections to water; …” Specify the method used for deparaffinizing.

Line 192, figure 1: add a scale bar on the image and specify at which stage each image corresponds.

Line 193, table 1: write “period” instead of “peroid”

Line 203, figure 2: add a scale bar to each figure.

Several captions on the photos would be useful: oocyte, granulosa, corpus luteum, corpus albicans, and other details that allow us to understand the microscopic anatomy of the ovaries at different stages of the cycle.

Author Response

  1. Line 38: use italics to write “Bos grunniens”.

Response 1:Thank you for your advice, we have revised according to your suggestion.

  1. Line 77: write “periods” instead of “peroids”.

Response 2:Thank you for your advice, we have revised according to your suggestion.

  1. Line 86: the authors write: “In this study, samples were collected from the ovaries of healthy adult yaks (6 years old), within one year and during three periods of anestrus (April), estrus (August) and pregnancy (December),…” Specify how many individuals were studied in each period.

Response 3:Thank you for your advice, in this study, ten ovarian samples were collected in each period for observing and making tissue sections, and three samples in each period were used for RNA-seq, it has been added in the text.

  1. Line 104: the authors write “Then the procedure consisted of: deparaffinizing sections to water; …” Specify the method used for deparaffinizing.

Response 4:Thank you for your advice, the specific steps of deparaffinizing are xylene I for 20 minutes, xylene II for 20 minutes, absolute ethanol I for 5 minutes, absolute ethanol II for 5 minutes, 75% alcohol for 5 minutes, and tap water rinsing, and it has been added in the text.

  1. Line 192, figure 1: add a scale bar on the image and specify at which stage each image corresponds.

Response 5:Thank you for your advice, we have added a scale bar on figure 1 and specified the corresponding stage for each image.

  1. Line 193, table 1: write “period” instead of “peroid”

Response 6:Thank you for your advice, we have revised according to your suggestion.

  1. Line 203, figure 2: add a scale bar to each figure.

Response 7:Thank you for your advice, we have added scale bar on figure 2.

  1. Several captions on the photos would be useful: oocyte, granulosa, corpus luteum, corpus albicans, and other details that allow us to understand the microscopic anatomy of the ovaries at different stages of the cycle.

Response 8:Thank you for your advice, we have supplemented figure 3 to introduce the microanatomic structure of secondary follicles in yak ovary in detail.
